

# To which extent are socio-hydrology studies really integrative? The case of natural hazards and disaster research

Franciele Maria Vanelli[1], Masato Kobiyama[1], Mariana Madruga de Brito[2]

[1]Institute of Hydraulic Research, Federal University of Rio Grande do Sul, Porto Alegre, 91501-970, Brazil
[2]Department of Urban and Environmental Sociology, Helmholtz Centre for Environmental Research, 04347, Leipzig, Germany

*Correspondence to*: Franciele Maria Vanelli (franciele.vanelli@ufrgs.br) Mariana M. de Brito (mariana.brito@ufz.de)

**Abstract.** Given the recent developments in socio-hydrology and its potential contributions to disaster risk reduction (DRR), we conducted a systematic literature review of socio-hydrological studies aiming to identify persisting gaps and discuss tractable approaches for tackling them. A total of 44 articles that address natural hazards or disasters were reviewed in detail. Our results indicated that: (i) 77.3% of the studies addressed floods whereas there were very few research applications for droughts (11.4%) and compound or multi-hazards (11.4%); (ii) none of the articles investigated interactions across temporal and spatial scales; (iii) quantitative approaches were used more often (65.9%) in comparison to mixed (22.7%) and qualitative (11.4%) approaches; (iv) monodisciplinary studies prevailed (61.4%) over multi or interdisciplinary (9.1%) ones, and (v) only 34.1% of the articles involved stakeholder participation. In summary, we found that there is a fragmentation in the field, with a multitude of social and physical components, methods and data sources being used. Based on these findings, we point out potential ways of tackling the identified challenges to advance socio-hydrology, including studying multiple hazards in a joint framework and exploiting new methods for integrating results from qualitative and quantitative analyses to leverage on the strengths of different fields of knowledge. Addressing these challenges will improve our understanding of human-water interactions to support DRR.

## 1 Introduction

In 2022, it will be one decade since Sivapalan et al. (2012) introduced the concept of socio-hydrology. It is also the last year of the scientific decade announced by the International Association of Hydrological Sciences (IAHS) entitled 'Panta Rhei – Everything Flows' (Montanari et al., 2013). Both initiatives are associated with the growing interest in understanding the interactions between society and water. Although research on human-water interactions is not a new subject (e.g. Falkenmark, 1977, 1979), these initiatives have been fruitful in engaging researchers (Madani and Shafiee-Jood, 2020). As a result, many socio-hydrological studies have been developed in recent years, including in the areas of floods (Di Baldassarre et al., 2013; Buarque et al., 2020), droughts (Kuil et al., 2016; Medeiros and Sivapalan, 2020), groundwater (Han et al., 2017; Herrera-Franco et al., 2020), and irrigation management (Sanderson et al., 2017), among others.



Within this context, socio-hydrology is often promoted as a key approach for integrating hydrological and social sciences perspectives with the aim of providing a holistic picture of complex systems. Socio-hydrology can deal with a range of policy-relevant questions concerning natural hazards, while hydrology alone cannot address these questions as it fails to consider how anthropogenic activities affect natural hazards and vice-versa (Di Baldassarre et al., 2021). In this context, socio-hydrology can support strategies to reduce negative impacts that are caused by interactions between societal vulnerabilities and natural

hazards (Di Baldassarre et al., 2018, 2021; Vanelli and Kobiyama, 2021). Nevertheless, although socio-hydrology can foster the integration of different knowledge types to understand coupled human-water systems, recent research has shown that the hydrologists' perspective still prevails (Seidl and Barthel, 2017; Xu et al., 2018). Reductionist and one-size-fits-all thinking is often used (Di Baldassarre et al., 2019). The consideration of these traditional (Hortonian) hydrology approaches departs from the Newtonian perspective of simplifying the complexity of nature to essential functions (McClain et al., 2012), whilst often

assuming that quantitative approaches (positivism) are superior to qualitative ones (interpretivism) (Seidl and Barthel, 2017). Nevertheless, some valuable insights on complex human-water relations cannot be quantified or addressed solely by traditional natural sciences tools (Di Baldassarre et al., 2021; Rangecroft et al., 2021). Hence, addressing human-water interactions requires wide interdisciplinary collaboration (Seidl and Barthel, 2017; Xu et al., 2018).

  The need for integrating different knowledge types is a crucial aspect of many socio-hydrology fields, especially for disaster

risk reduction (DRR). Indeed, both scientific and local knowledge are required to mitigate risk and reduce the negative impacts of disasters in a comprehensive way (Rai et al., 2011; Vanelli and Kobiyama, 2021). In this regard, the Sendai Framework claims that DDR should be based on an understanding of disaster risk in all its dimensions of vulnerability, capacity, exposure, hazard characteristics, and the environment. Hence, building disaster resilience and reducing losses requires an integrative approach and all-of-society engagement and partnership (UNDRR, 2015). Similarly, the recent report of the Intergovernmental

Panel on Climate Change (IPCC, 2021) calls for multidisciplinary and transdisciplinary groups because risks can arise not only from the impacts of climate change but also from human responses to it.

  Studying the complexity of natural hazards and their interactions with society thus requires us to overcome current dichotomous ways of thinking, i.e. natural sciences × social sciences; researchers × stakeholders; and quantitative × qualitative. These can be addressed by integrating different disciplines (interdisciplinary), science and society (transdisciplinary), and

quantitative and qualitative data and methods (mixed methods). The use of integrative approaches does not mean a 'homogeneity' of the parts but it considers that each perspective is relevant and has advantages and disadvantages. Working with the pluralism of philosophies, methodologies, backgrounds, and experiences is challenging but it can provide new ideas, understandings, and potential solutions for complex problems (Krueger et al., 2016; Rangecroft et al., 2021; Slater and Robinson, 2020).

The main objective of this study was to investigate the current state of the art regarding socio-hydrological studies in the areas of natural hazards and disasters. Our research particularly focused on evaluating the extent to which current applications are holistic in terms of considering coupled physical and social systems, as well as how they integrate different types of knowledge. To this end, a systematic literature review was conducted. The following questions guided the analysis: (i) Which disasters





triggered by natural hazards are addressed in socio-hydrology studies?; (ii) How are these studies distributed across different

countries and continents?; (iii) Which spatial and temporal scales are considered?; (iv) How are coupled social and physical (natural) systems illustrated or represented?; (v) Which methods are used to gather and process data?; and (vi) To which extent are these studies inter and/or transdisciplinary? By answering these questions, we synthesised current research, identified persisting gaps, and provided possible ways forward.

## 2 Methods

To ensure objectivity, transparency, and reproducibility, the systematic review followed the ROSES reporting standards (Haddaway et al., 2018). Searches were performed on 12 February 2021 on the Web of Science (WoS) and Scopus databases. No start time constraints were used; however, only articles published before 31 December 2020 were considered. We searched for the following search terms in the title, abstract, and author's keywords: ("socio-hydrology" OR "sociohydrology" OR "socio-hydrological" OR "sociohydrological" OR "socio-hydrologic" OR "sociohydrologic"). In addition, both databases were

searched for terms related to hydrology in general: ("hydrology" OR "hydrological" OR "hydrologic"). This allowed us to normalise the data, so that we could correctly measure the temporal trend in the number of socio-hydrology publications. The queries were restricted to English-language and peer-reviewed articles. See Supplementary Table S1 for more detail on the search strings used.

The review process involved a set of progressive steps (Figure 1). At first, duplicate articles (n=189) were removed from the

sample. The 231 remaining articles were screened according to the eligibility criteria shown in Table 1: first at the title and abstract level, and then on the full-text level. A total of 54 articles were retrieved for full-text analysis. Of these, five were disregarded as they were reviews, editorials, or opinion articles: Di Baldassarre et al. (2018), Borga et al. (2019), Gober and Wheater (2015), Wens et al. (2019), and Westerberg et al. (2017). A further five articles were removed from the analysis as they did not address social system components. Here, 'social systems' correspond to individuals, groups, institutions and their

interactions, whereas 'physical systems' refer to physical entities, processes and their relationships. After the screening step, 44 articles were deemed relevant to the objectives of our review.

In order to answer the research questions, the articles were categorised according to the following criteria: (a) country of the study, (b) type of natural hazard investigated, (c) spatial scale of the social and physical systems, (d) temporal scale of the social and physical systems, (e) physical and social components, (f) social and physical data gathering sources, (g) social and

physical data processing methods, (h) methodological approach type, and (i) inter and transdisciplinarity. The classification followed an inductive reasoning approach and was conducted by the first author (FMV). Uncertainties were resolved through discussion between the reviewers (FMV and MMdB).

To classify the articles' spatial and temporal scales, we considered the scales indicated by the authors in the methods or results sections. Likewise, to classify the articles' methodological approaches, we considered the data gathering and processing

techniques mentioned in the text. For instance, if an article used quantitative techniques to gather and process data, we





classified it as a quantitative approach; if qualitative techniques were used, we classified it as a qualitative approach. Articles that used a mixed research design, such as qualitative techniques for data gathering (e.g. narratives or focus groups) and quantitative techniques for data processing (e.g. agent-based modelling or statistical analysis), were added to a mixed approach category. Mixed approaches are also referred to as integrative research, mixed methods research, multiple research,

triangulation, and multi-strategy, among others (e.g. Di Baldassarre et al., 2021; Bryman, 2007; Creswell, 2012; Johnson and Onwuegbuzie, 2004; O'Cathain et al., 2010). In cases where the techniques used by the authors were not clear, we considered how the results were presented and analysed (qualitative or quantitative).

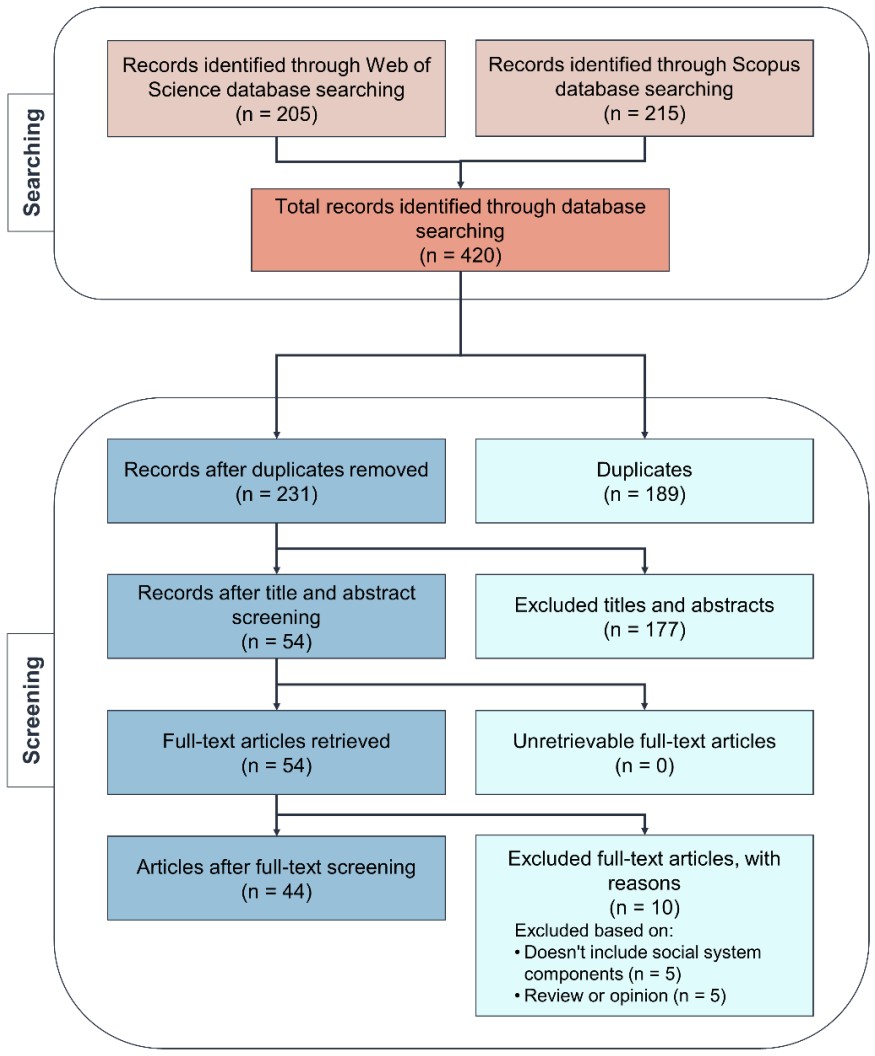

**Figure 1: Process used to select relevant articles adapted from the ROSES flow diagram for systematic reviews. Note that 'n'**
**indicates the number of articles in each step or item.**

The interdisciplinarity of each article was classified as 'monodisciplinary', 'weakly inter or multidisciplinary', or 'inter or multidisciplinary'. To do this, we followed previous studies (Seidl and Barthel, 2017; Xu et al., 2018) and considered the





composition of the authors' disciplinary perspectives (natural or social sciences), based on their affiliations. Conversely, transdisciplinarity was classified in a binary way and was defined as studies that go beyond disciplinary boundaries and include

stakeholders in the research design. To this end, we verified if stakeholders outside academia such as citizens, decision-makers, and policymakers were included in the study at a certain level (e.g. through filling out surveys or attending focus group discussions, among other activities).

**Table 1: Eligibility criteria used to identify which studies should be included in the review.**

| Inclusion criteria | Exclusion criteria |
|---|---|
| ▪ Focus on natural hazards, risks or disasters (e.g. flood, drought, landslide, earthquake, among others) | ▪ Focus on water quality, virtual water, agriculture, groundwater, sustainability index, or water consumption |
| ▪ Articles acknowledging interactions between social and physical systems | ▪ Assessments that do not apply social components |
| ▪ Case studies (real or hypothetical) | ▪ Reviews, editorials or opinion articles |

## 3 Results and discussion

### 3.1 Overview of the socio-hydrology articles

Even though some articles mentioned the term "socio-hydrology" before 2012 (e.g. Lele, 2009; Mohorjy, 1989), the first article to propose the simultaneous investigation of both social and physical components was published in 2012. To evaluate the temporal publication trends, the number of "socio-hydrology" articles published between 2012 and 2020 was normalised by the number of hydrology studies per year (Figure 2). The results show a growing number of socio-hydrology articles in the

last years, with most of those articles published in 2020. This growth can be explained by discussions on socio-hydrology prompted by key journals in the field. In 2015 Water Resources Research published an editorial called "*Debates – Perspectives on Socio-Hydrology*" (which is reflected by the first peak in article numbers) and between 2017 and 2018, a special issue titled "*Socio-hydrology: Spatial And Temporal Dynamics of Coupled Human-Water Systems*". The Journal of Hydrology organised the "*Virtual Special Issue on Building Socio-hydrological Resilience*" between 2018 and 2019. Meanwhile, the Hydrological

Sciences Journal published articles in its "*Virtual Special Issue: Advancing socio-hydrology: a synthesis of coupled human–water systems across disciplines*" between 2019 and 2020. It is worth mentioning that these special issues focused on socio-hydrology in general. None of them focused exclusively on disasters triggered by natural hazards.

After the screening step, a total of 44 articles were deemed relevant and reviewed in detail. Of these, 59% cited Sivapalan et al. (2012) when defining the concept of socio-hydrology, 41% did not provide a definition, and none presented a new or revised

concept. Hence, although researchers have used the term "socio-hydrology" more frequently during recent years, the predominance of the definition by Sivapalan et al. (2012) indicates that socio-hydrology is still a developing field.

The articles' spatial distribution (Figure 3) shows that the studies are concentrated in just a few countries, namely Italy (n = 7) and Bangladesh (n = 6). This result is surprising because the number of studies does not correspond to the countries with the highest number of disasters between 2000 and 2019 (United Nations for Disaster Risk Reduction, 2020). Furthermore, we did

not identify many articles from the United States, Australia and China, even though those countries generated a high frequency
of studies according to a recent review on socio-hydrology by Fischer et al. (2021).

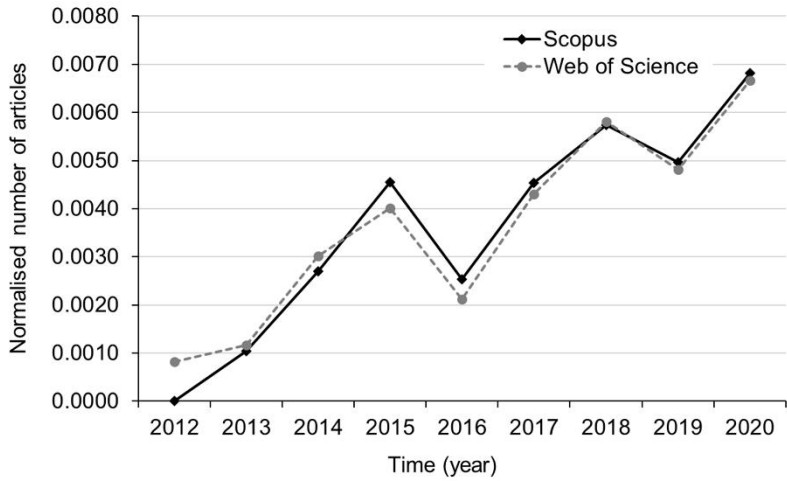

**Figure 2: Number of socio-hydrology articles normalised by the number of hydrology ones published between 2012 and 2020 based
on data from the Web of Science (214 socio-hydrology articles and 53,175 hydrology ones) and Scopus (204 socio-hydrology articles
and 50,785 hydrology ones) databases.**

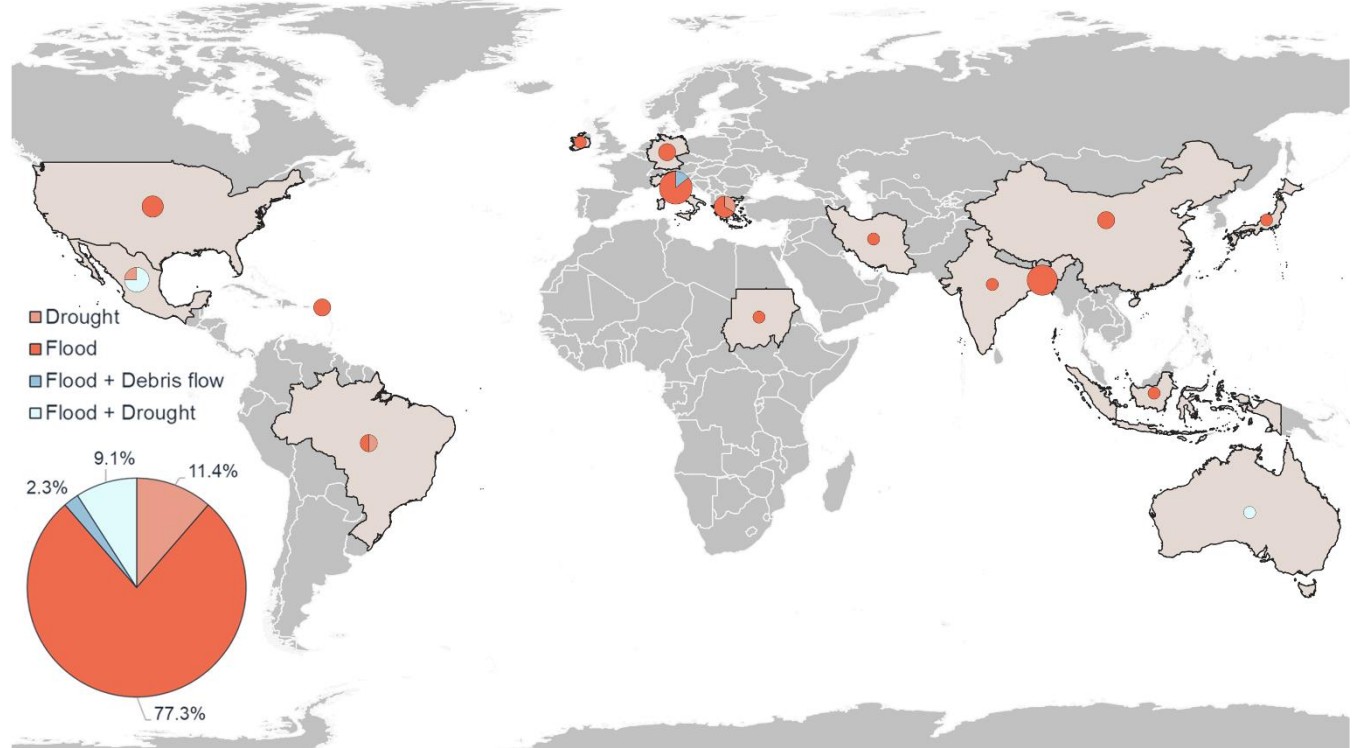

**Figure 3: Spatial distribution of the study areas and hazards investigated. The size of the circle is proportional to the number of
articles. Hypothetical case studies (n=6) and studies about the Maya civilisation (n=2) are not represented on the map.**





Concerning the type of natural hazard investigated, the results show that the studies focused predominantly on floods (77.3%).
Few studies investigated droughts (11.4%) and multi-hazards (11.4%) (pie chart in Figure 3). This is concerning, as droughts
cause impacts of similar magnitude as floods. Indeed, from 1900 to 2018 floods caused 4.4 million fatalities, affected 2.5
billion people. Droughts, on the other hand, killed 11.7 million persons and affected 2.7 billion people (EM-Dat, 2021). Most
articles did not detail the type of flood or drought investigated. Of the 34 flood articles, only 26.5% presented details about the
type of flood studied: flash flood (5.9%), coastal flood (5.9%), urban and coastal flood (2.9%), pluvial and coastal flood (2.9%),
riverine and coastal flood (2.9%), flash and riverine flood (2.9%), and urban flood (2.9%). None of the articles specified the
type of droughts studied (i.e. meteorological drought, hydrological drought, agricultural drought). This is particularly relevant
as different drought types have different implications for their management (Hagenlocher et al., 2019). Regarding multi-hazard
studies, most of the articles that investigated droughts also addressed floods. Indeed, 9.1% of the 44 articles studied floods and
droughts (Albertini et al., 2020; Baeza et al., 2019; Lerner et al., 2018; Shelton et al., 2018). Moreover, Mondino et al. (2020a)
analysed flood and debris flow occurring as compound hazards. Other types of natural hazards, such as earthquakes and
heatwaves, where water is an indirect trigger and/or essential for the disaster response (Vanelli and Kobiyama, 2021), were
not identified.

## 3.2 Trends regarding the studies' spatial scale

Different spatial scales have been used in socio-hydrological studies, with the majority (86.4%) considering distinctive spatial
scales between social and physical systems (Figure 4). Indeed, there are clear differences between the scales used for
characterizing each system ($p= 0.00049$, Fisher's exact test). Some studies applied more than one spatial scale for social (6.8%)
or physical (2.3%) systems. These studies were defined as "Multiple scales" because the presence of more than one scale did
not imply interactions across different scales (cross-scale). In fact, none of the reviewed articles conducted cross-scale analyses
where the result of processes at one scale interacted with other processes at another scale (Soranno et al., 2014).
For the social systems, there was a preference for detailed scales. Even though there is no convention concerning which spatial
scale can provide a better overview of social processes, socio-hydrology studies often need detailed information on the exposed
people and communities. As such, "Individual or Household" and "Group or Community" were the most used spatial scales
(36.2%). This is similar to the findings of Moreira et al. (2021), who found that flood vulnerability studies tend to focus on the
neighbourhood scale due to data availability. Studies that focus on the individual level are also popular as they enable the
collection of specific behavioural information. Among the actors involved in these studies, it is important to highlight
companies (Grames et al., 2019), government agents (Abebe et al., 2019), one-person households (Mondino et al., 2020a),
local communities, stakeholders, and researchers (Maghsood et al., 2019). Few studies used political units as the spatial scale:
"Municipal" (8.5%), "Regional" (4.3%), and "National" (8.5%). This is surprising, as public policies and laws for DRR are
often defined by considering political boundaries. No studies were conducted on the "Global" scale and only 4.3% of the
studies used the basic unit of hydrology, the "River basin", to characterise social systems.





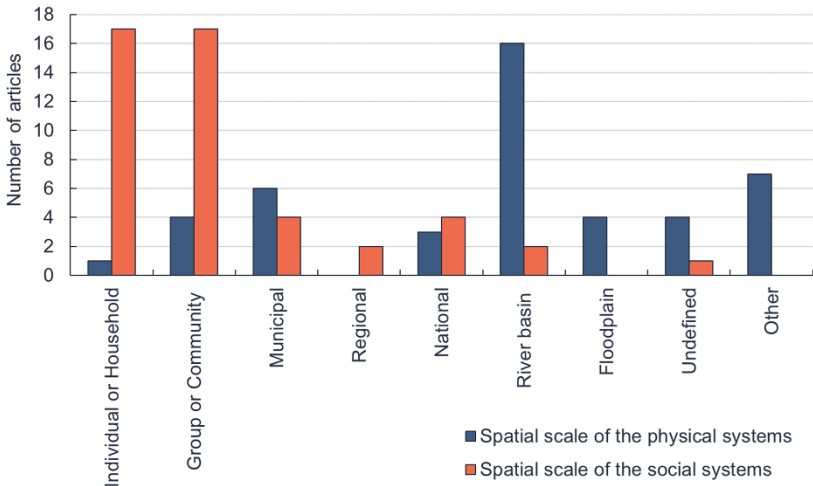

**Figure 4: Spatial scales used to characterise the physical and social systems in socio-hydrology studies. The total number of articles is higher than 44 as some articles used more than one scale.**

For physical systems, 35.6% of the studies used "River basin" as the spatial scale. This was expected, as the basin scale is the
conventional scale used for hydrological analysis. Furthermore, as indicated previously, most of the reviewed studies dealt
with floods (Figure 3). The use of political units was infrequent: "Municipal" (13.3%), "National" (6.7%). There were no
studies on "Regional" or "Global" scales. Of the 44 articles, only 2.2% used "Individual or Household" as the spatial scale for
physical systems and 8.9% used "Group or Community". Besides this, 15.6% of the studies relied on other spatial units:
engineering structures, such as dams (Wallington and Cai, 2020) or polders (Sung et al., 2018; Yu et al., 2017), physical
delimitations like groundwater (Basel et al., 2020), and floodplains (Ferdous et al., 2018, 2020; Han et al., 2020; Wang et al.,
2020). One study proposed a new spatial unit: the 'socio-hydrological systems boundaries', which is defined by the outer
contour of the river basins and the water supply networks (Sapountzaki and Daskalakis, 2016).

Differences were observed when comparing the scales used in different continents (Figure 5). The traditional and common
unit of hydrology, the "River basin", was the scale most used to address physical systems in Europe, Oceania, and South
America (Figure 5a). Asian studies predominantly used "Floodplain" as the spatial scale of physical systems. For instance,
Wang et al. (2020) used the "Floodplain" and defined it as the flood hazard extent with a 100-year return period. Meanwhile,
Ferdous et al. (2018, 2020) and Han et al. (2020) defined the "Floodplain" extent based on previous flood events. For the social
system, in all continents except Africa, most studies were conducted at the level of individuals or small groups (Figure 5b).



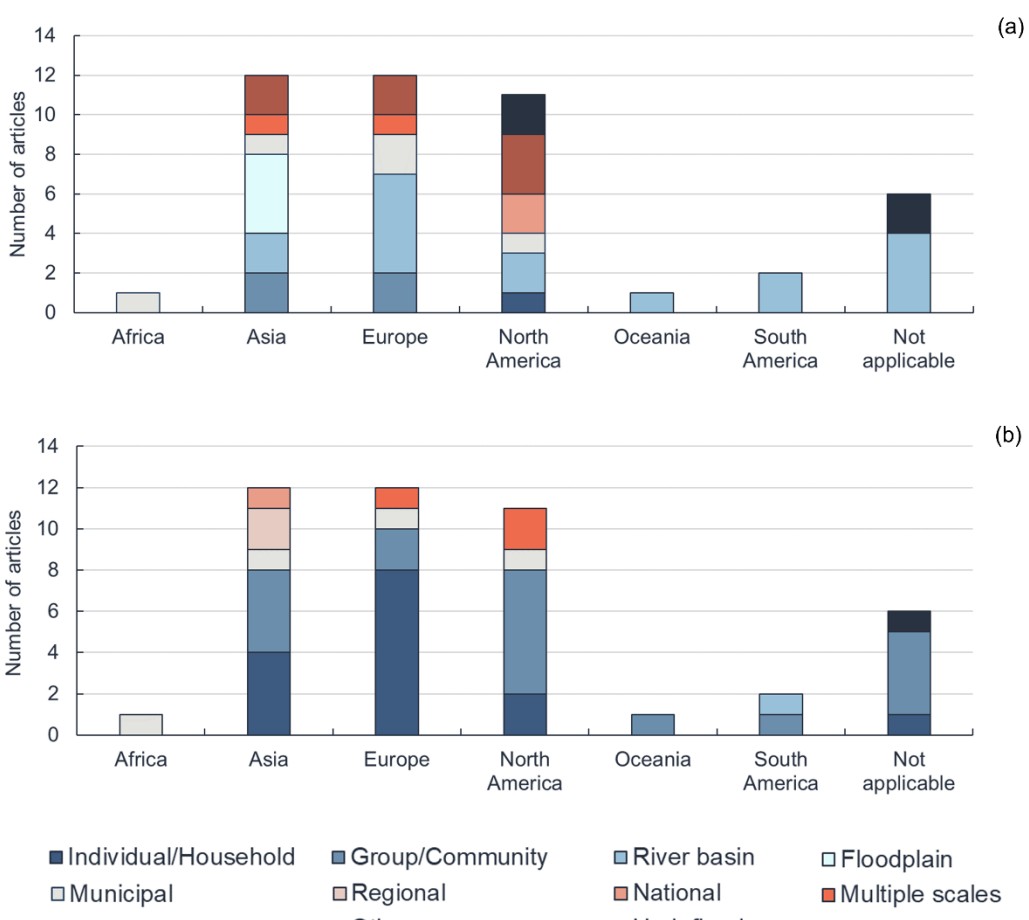

**Figure 5: Number of articles by continent and spatial scale of the (a) physical system and the (b) social system. Hypothetical case studies were indicated as "Not applicable" and studies about the Maya civilisation were attributed to North America.**

### 3.3 Trends regarding the studies' temporal scale

The variation witnessed with regard to the spatial scales is not evident in relation to the temporal scale. Most studies (72.7%) used the same temporal scale to address both physical and social systems (Figure 6). For physical systems, the analyses were predominantly associated with a one-time "Extreme disaster event" (29.5%), followed by a "Yearly" perspective (27.3%). The "Yearly" scale predominated (50%) for social systems and only 4.5% of the studies investigated one-time processes by considering the "Post-disaster event" scale. Usually, the yearly temporal scale of social systems was associated with the application of data-intensive tools, such as mathematical modelling or agent-based modelling. Within this context, the unavailability of a temporal series of social data is a notable gap.

Some studies were classified as "Other" as they compared different temporal periods (Han et al., 2020; Nakamura and Oki, 2018) or conducted longitudinal surveys in different years (Mondino et al., 2020a). In another example, Shelton et al. (2018)





considered four years and four rounds of analyses per year—two rounds for the dry season and two for the wet season. The temporal scale was not clearly defined in 20.5% of the studies.

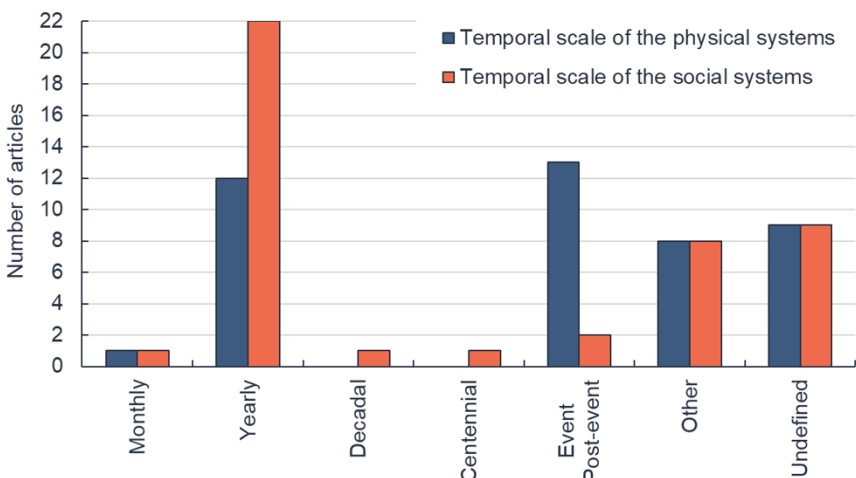

**Figure 6: Temporal scales used to characterise the physical and social systems in socio-hydrology studies related to natural hazards and disasters.**

### 3.4 Trends regarding the social and physical components of coupled systems

To understand how coupled social and physical systems were illustrated or represented in the analysed studies, we classified the articles according to their respective components. Figure 7 shows the (a) physical and (b) social components according to the hazard investigated. Most studies (56.8%) used more than one social component, while 25% simplified the physical system by considering only one component. For floods, an average of 1.4 physical components and 2.4 social components was used per study, whereas for droughts the average was 1.2 for the physical components and 2.0 for the social components. Although many studies used similar components, there was a lot of variety, which is an indicative of the vitality of the field. For instance, all four articles that analysed floods and droughts simultaneously applied different combinations of components (Albertini et al., 2020; Baeza et al., 2019; Lerner et al., 2018; Shelton et al., 2018). Often, the system components were not described in detail, thereby hampering the reproducibility of the results.

For physical systems, water level—categorised here as a "Hydraulic" component—was used in 47.1% of the studies that address floods (e.g. Di Baldassarre et al., 2017; Ciullo et al., 2017; Viglione et al., 2014). For droughts, 60% of the studies applied a "Hydrological" component (Basel et al., 2020; Kuil et al., 2016, 2019). Several articles (38.6%) used other physical components, such as tide level (Sung et al., 2018; Yu et al., 2017) or hazard (Ferdous et al., 2018, 2020; Leong, 2018; Mondino et al., 2020b, 2020a). Some studies (13.6%) did not identify the physical components in detail and were classified as "Undefined".

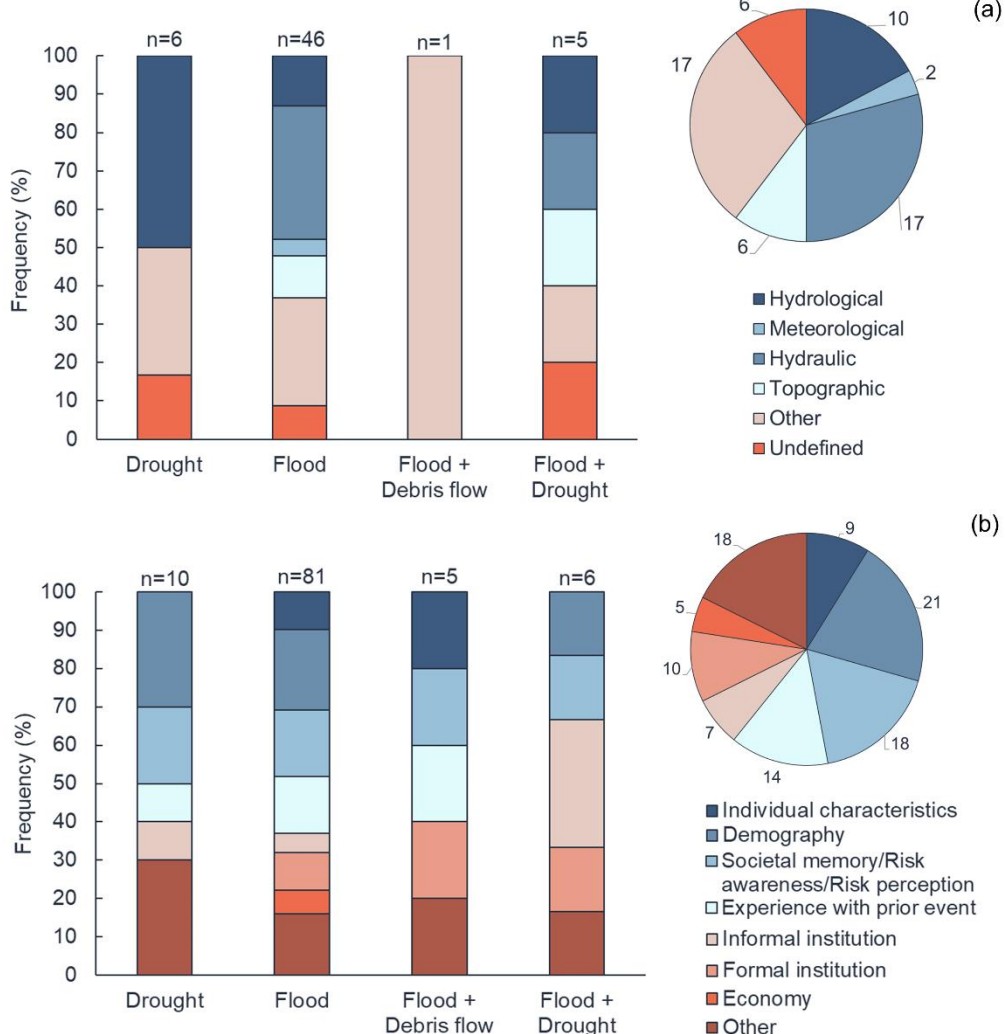

**Figure 7: (a) Physical components and (b) social components related to the type of natural hazard. The sum is not equal to 44 as most articles used more than one component.**

Among the 54 articles that had their full text screened (Figure 1), five were excluded as the social components were not considered. This indicates that the "social" in socio-hydrology has not yet been clearly defined, with divergences among authors about concepts and applications. This ambiguity in the definition of social system components was also identified by Fischer et al. (2021) in their review of socio-hydrology studies. Conversely, the physical components such as discharge, measured in $m^3 \cdot s^{-1}$, or water level, measured in $m$, and their causal relations are more standardised. Hence, here we will focus on the social components, aiming to provide an overview of the aspects often considered in socio-hydrological studies related to natural hazards and disasters.





Overall, the most used social component was "Demography" (47.7%), which is used to understand people's exposure to natural
hazards and the potential for population displacement. "Demography" was used in mathematical modelling (stylised model)
to calibrate "Societal memory" (e.g. Albertini et al., 2020; Di Baldassarre et al., 2013; Buarque et al., 2020). The "Informal
institutions" component refers to community behaviour like local norms, rules, or attitudes, and it was used in 15.9% of the
studies. Meanwhile, "Formal institutions" (22.7%) are based on policies, laws, or norms. "Experience with prior events" was
used mainly in flood studies (27.3%), and it was associated with the magnitude of psychological shock experienced (e.g. Di
Baldassarre et al., 2013, 2017; Buarque et al., 2020; Viglione et al., 2014). In a drought study, Sapountzaki and Daskalakis
(2016) investigated people's experiences with previous droughts by using structured questionnaires. In a study about flood and
debris flow, Mondino et al. (2020a) used "Experience", "Risk awareness", "Formal institution", and "Individual
characteristics" as social components.

The only social component that was applied in conjunction with all the identified types of natural hazards was "Societal
memory/Risk awareness/Risk perception". These concepts were grouped together as they are used synonymously by many
authors (e.g. Michaelis et al., 2020; Sawada and Hanazaki, 2020). However, it should be highlighted that although risk
awareness and risk perceptions are correlated, they are not interchangeable (Mondino et al., 2020b). Hence, despite 40.9% of
the studies applying this category to illustrate or represent the social system, there is a gap in the consolidation of these
concepts.

The social component "Economy" was used by 11.4% of the articles. For instance, Abadie et al. (2019) framed socio-hydrology
as an optimisation problem and included an economic valuation of costs and benefits, such as the rent on occupied land, and
the costs of increasing and replacing flood defences. Sung et al. (2018) and Yu et al. (2017) considered the total annual income
of a household in their socio-hydrological model. Grames et al. (2016) introduced an optimal decision framework to investigate
the interaction between a society's investment in flood defence and its productive capital. In a subsequent study, Grames et al.
(2019) focused on corporate decisions to invest in flood protection. In some studies, like Di Baldassarre et al. (2013), the
economy was mentioned, but it was related to the growing or shrinking of human settlements in response to flooding. Such
studies were therefore only included in the "Demography" category.

Many studies used "Other" social components besides the categories described above. For instance, Kuil et al. (2016)
investigated the droughts that affected the Maya civilisation and applied vulnerability as a social component in addition to
memory and demography. Similarly, Chen et al. (2016) applied community sensitivity for studying floods in the USA, in
which a higher value of sensitivity represents a greater tendency of the community to take actions favouring the environment.

## 3.5 Trends regarding the methods used to understand coupled social and physical systems

To investigate the most common methods used, we classified the articles according to the data gathering source and processing
techniques used. It is important to highlight that we only considered the sources and methods that were explicitly mentioned
by the authors. More than 35% of the 44 reviewed studies did not specify the sources used for gathering physical data and 25%
did not specify their social data sources (see the "Undefined" category in Figure 8a). Of the 44 studies, 34.1% used more than





one data source for social components. For physical components this figure is 15.9%. Although some studies used "Interviews" and "Focus Groups" to collect physical data, this type of data was primarily sourced from "Gauges" and "Remote sensing". Conversely, "Questionnaires" were the primary sources for social data gathering, followed by "Census data" and "Official
documents".

With regard to the data processing tools, mathematical modelling (i.e. stylised model) was the most used technique for both social and physical systems (Figure 8b). This quantitative technique uses differential equations to represent the system; the resulting modelling has less detail but is intended to capture the system's holistic aspects in a general way (Sivapalan and Blöschl, 2015). An example of a socio-hydrological stylised model is provided by Di Baldassarre et al. (2015). It is a
simplification of a previous model (Di Baldassarre et al., 2013; Viglione et al., 2014) that was applied in several studies (Di Baldassarre et al., 2017; Buarque et al., 2020; Ciullo et al., 2017; Sawada and Hanazaki, 2020) and was also modified by Abadie et al. (2019), who included the economy as a component.

"Statistical analysis" and "Agent-based modelling" were used to process social systems data in 18.2% and 15.9% of the studies, respectively. Few studies applied more than one technique for processing social data (Horn and Elagib, 2018; Maghsood et
al., 2019; Mondino et al., 2020a; Sapountzaki and Daskalakis, 2018), and only Sawada and Hanazaki (2020) applied two techniques (mathematical modelling and data assimilation) for both social and physical data. Among the "Other" techniques, for instance, Leong (2018) applied the quantitative Q methodology to study the subjectivities that explain how the social memory of floods results in different vulnerability or adaptive responses. Only Sugeng et al. (2019) applied "System dynamics"; they created causal loop diagrams using the Vensim software.

Several data sources and techniques were used for assessing the same component. Hence, when we conducted a cluster analysis, it was not possible to identify patterns or trends in the components and methods used (Supplementary Figure 1). In one example of this diversity, Nakamura and Oki (2018) considered flood hazard and formal institutions data gathered from official documents and processed them by means of a content analysis. Koutiva et al. (2020) used similar components, but the data were obtained through "Questionnaires" and "Focus groups" and processed using an "Agent-based model". This
fragmentation and lack of guidelines about which data and methods should be used makes it difficult to compare the studies.

The classification of the data gathering sources and processing methods (Figure 8) indicated the predominance of quantitative approaches (65.9%) over mixed (22.7%) and qualitative (11.4%) ones. Although socio-hydrology was originally proposed as a quantitative field (Sivapalan et al., 2012), there are limitations to studying the human-water system solely using quantitative data and methods (Wilson et al., 2015). Our results revealed the limited integration of quantitative and qualitative data. Indeed,
data analyses were often carried out separately and their findings were not combined. However mixed approaches can make a valuable contribution to a holistic understanding of interwoven social and environmental processes. Only a few of the reviewed studies used mixed approaches, such as the translation of qualitative data in agent-based models (Shelton et al., 2018) or data triangulation (Ferdous et al., 2018).

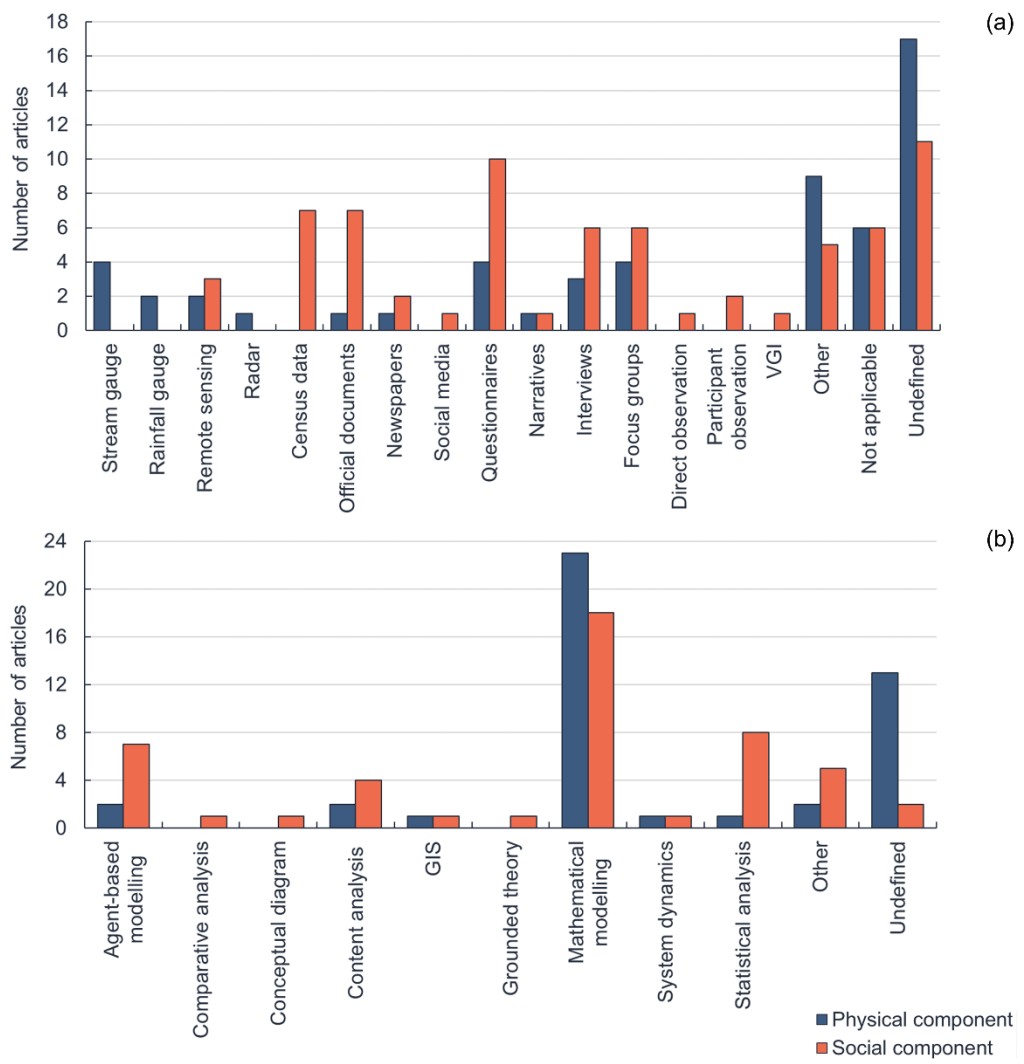

Figure 8: (a) Data gathering and (b) data processing tools used for understanding the physical and social systems. The sum is not equal to 44 as most articles used more than one data source or processing technique. "Not applicable" refers to studies where data was not gathered, as they address hypothetical cases.

### 3.6 Trends regarding the studies' inter and transdisciplinarity

Although disasters involve social and natural aspects, and consequently, different researchers and techniques, our review demonstrates the predominance of monodisciplinary studies (61.4%)—mainly from natural sciences (Figure 9a). This finding is in line with those of Seidl and Barthel (2017) and Xu et al. (2018): socio-hydrology is still dominated by hydrologists who have adopted a hegemonic attitude toward interdisciplinary collaboration with social scientists. Among the multi or interdisciplinary studies, the working groups often involved hydrologists, physical geographers, social scientists, economists, mathematicians, and ecologists (e.g. Abadie et al., 2019; Baeza et al., 2019; Mondino et al., 2020a).





Besides the infrequent use of iterative approaches among social and natural scientists, few monodisciplinary studies included stakeholder participation (33.3%) compared to multi or interdisciplinary studies, where stakeholder participation dominated (75%) (Figure 9b). This can lead to results that are not trusted—and therefore not used—by stakeholders, since they were not involved in the analysis (de Brito et al., 2017; Evers et al., 2018). In the few transdisciplinary studies we reviewed, the participation of stakeholders occurred at different levels. For instance, Mondino et al. (2020a) gathered longitudinal social data

through more than 450 questionnaires in two communities. Ferdous et al. (2018) collected approximately 900 questionnaires and conducted 12 focus group discussions, each with 20 participants. Koutiva et al. (2020) used the results of questionnaires and workshops to design a model. One article (Basel et al. 2020) was even co-written by local leaders in collaboration with researchers.

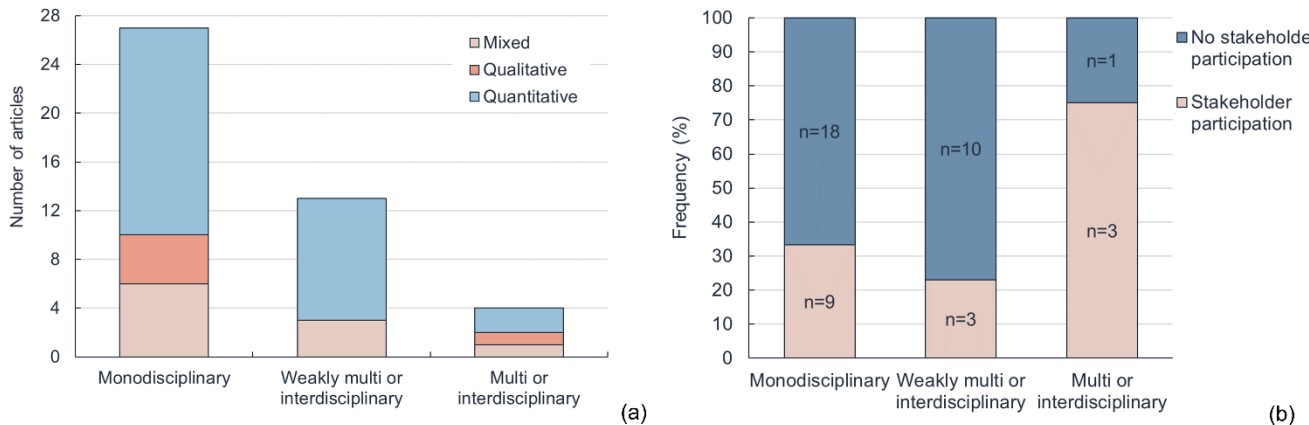

**Figure 9: Approach identified in the studies at (a) the interdisciplinary level and (b) the transdisciplinary level.**

## 4 Research agenda

In the last five years, there has been an upsurge in the number of socio-hydrology studies applied to natural hazards and disasters. By systematically reviewing 44 studies, we found that considerable achievements have been made. However, our results also underlined that current knowledge is limited with respect to several key areas. In this section, we summarise the

challenges we have identified and propose an agenda for future research (Table 2).

The first persisting knowledge gap is related to the predominance of flood studies. Even though droughts and multi-hazard events cause comparable damages as floods they received little attention. Yet, the occurrence of compound and cascading natural hazards is expected to increase in the future (AghaKouchak et al., 2018, 2020; De Brito, 2021). Besides this, all types of disasters can be considered as socio-hydrological phenomena as they are directly and/or indirectly associated with water

(Vanelli and Kobiyama, 2021). Hence it is necessary to advance our understanding of interwoven social and environmental processes by considering the interplay between society and different types and combinations of natural hazards. To this end, new methods and data are needed to consider the dynamics between consecutive and compound hazards and society.



A second gap refers to the lack of cross-scale analyses. An investigation of the interplay across spatial scales is often advocated by socio-hydrologists (e.g. Di Baldassarre et al., 2019; Pande and Sivapalan, 2017; Sivapalan et al., 2014). This is required as

the dynamic interactions between natural hazards and society share nonlinear behaviours that are driven by forces interacting across spatial and temporal scales (Adger et al., 2005; Birkmann and von Teichman, 2010; Nelson et al., 2006; Peters et al., 2004; Räsänen, 2021; Vanelli and Kobiyama, 2021). Therefore, recognising the interactions across scales is fundamental for improving future projections, particularly in systems dominated by changing social dynamics (Srinivasan et al., 2017; York et al., 2019). Yet, no cross-scale studies were identified in our review. We observed an overreliance on local case studies that

ignore broad socio-political contexts and vice-versa. The use of static scale analysis, thus, limits our ability to understand cross-scale connections, which can in turn lead to maladaptive practices (Ford et al., 2018). In this context, the concept of "glocal" proposed by Robertson (1994) and Swyngedouw (2004) can be used to strengthen the idea of interactions and feedbacks occurring across different scales, where global connections influence the local level while local heterogeneous characteristics simultaneously influence global strategies. However, setting up such analyses is challenging due to the lack of

data on human-water interactions (Brunner et al., 2021).

A third gap refers to the wide range of understandings of what 'social' means in socio-hydrology. Several articles were removed during the screening stage because, even though the authors stated that they had conducted a socio-hydrological study, no social aspects were actually considered. As Basel et al. (2020) have pointed out, socio-hydrology is still developing knowledge of the variables that drives the coupled system. As such, concepts and interpretations of which social components

should be considered remains contested. Such confusion makes it difficult to draw comparisons between the studies and complicates the production of cumulative insights and the identification of patterns among multiple studies (i.e. by conducting meta analyses). Hence there is a need for a deeper understanding of the social components and their causal relations. We suggest that, rather than striving for a unified approach to address the social components in socio-hydrology studies, scientists should be explicit about the variables they use and the reasons for doing so.

The fourth gap concerns the predominance of quantitative approaches for gathering and processing data. The use of mixed-methods research designs make it possible to better understand the diverse social, economic, environmental, and political parts that make up natural hazards and disasters (Eriksen et al., 2011). Using different data or methods to test a hypothesis is an effective way to check its validity and reliability (Jick, 1979), because when different methods produce the same or similar results they are less likely to be artefacts (Munafò and Davey Smith, 2018). Mixed-methods approaches can enhance our

confidence in the findings and be used to assess whether data agree (convergence), complement one another (complementarity) or contradict each other (O'Cathain et al., 2010). Hence, the integration of qualitative and quantitative data and methods should be used in future studies to examine socio-hydrological phenomena from multiple perspectives, as this allows us to expand or deepen our understanding of the social components in the coupled system (Di Baldassarre et al., 2021; Vanelli and Kobiyama, 2021; Wilson et al., 2015). To this end, different types of mixed research designs can be used, including simultaneous

quantitative and qualitative data collection and analysis (a convergent parallel design) or the sequential collection and analysis of data (explanatory sequential design) (Creswell, 2012).





**Table 2: Research agenda for socio-hydrological studies involving natural hazards and disasters.**

| Main gaps | Research needs | Examples of ways forward |
|---|---|---|
| ▪ To broaden the application of socio-hydrology to other natural hazards and disasters, beyond flood studies | ▪ Investigate droughts and compound or cascading disasters (e.g. floods and landslides or heatwaves and soil moisture droughts that trigger wildfires) | ▪ Hydrogeological hazards (Mondino et al., 2020a), floods, and droughts (Albertini et al., 2020) |
| ▪ To consider human-water interactions across temporal and spatial scales | ▪ Conduct cross-scale studies to understand interactions between social and physical components across spatial and temporal scales | For socio-hydrology in general:<br>▪ Cross-scale interactions in socio-hydrological subsystems used to assess water resource management (York et al., 2019)<br>For disaster research in general:<br>▪ Cross-scale interactions in flood risk management (Räsänen, 2021)<br>▪ Cross-scale interactions in vulnerability and resilience assessment (Gotham and Campanella, 2011) |
| ▪ To strengthen the 'social' in the socio-hydrological study of coupled human-water systems | ▪ Clear definition of the social components considered<br>▪ Investigate causal relations between social and physical components<br>▪ Give the social components the same importance as the physical ones | ▪ Definition of relevant social variables for understanding local processes (Basel et al., 2020; Mondino et al., 2020a) |
| ▪ To broaden the methodological repertoire in socio-hydrology studies | ▪ Use of mixed approaches | ▪ Translation of qualitative data in agent-based models (Shelton et al., 2018)<br>▪ Data triangulation (Ferdous et al., 2018) |
| ▪ To catalyse collaboration across disciplines and stakeholders | ▪ More interdisciplinary working groups<br>▪ Genuine involvement of the stakeholders in all stages of the study | ▪ Team of hydrologists, social scientists, and physical geographers (Mondino et al., 2020a)<br>▪ Stakeholders contributed to the writing of the article (Basel et al., 2020)<br>▪ Stakeholders contributed to the model design (Koutiva et al., 2020) |
| ▪ To consider research ethics principles | ▪ Define clear guidelines and rules to foster ethical and equitable relationships<br>▪ Use techniques to protect privacy | ▪ Guidelines for improving social data gathering (Rangecroft et al., 2021)<br>▪ Aggregation of data to a level at which no individual is identifiable (Flint et al., 2017)<br>▪ The use of proper geomasking methods (Kounadi and Leitner, 2014) |

The fifth gap refers to the low frequency of inter and transdisciplinary studies among social and natural scientists, as well as

among scientists and stakeholders. Although the study of natural hazards and disasters is interdisciplinary, most of the studies



we reviewed were monodisciplinary, conducted by hydrologists, and with low stakeholder participation. This raises the question: if socio-hydrology uses the same methods and perspectives as traditional hydrology, can we expect it to deliver different and new insights into complex human-water systems? Collaborative discussions and research between the social and natural sciences can significantly enhance the way research is designed and carried out, as well as produce holistic outputs

(Carr et al., 2020; Rangecroft et al., 2021; Thaler, 2021). By engaging in a dialogue with key players and decision-makers, we can design models and solutions that address users' needs. Furthermore, transdisciplinary development helps to improve the sense of plural perspectives, to transform empirical knowledge into actionable knowledge, and, particularly for DRR, to enhance the credibility and deployment of results (De Brito et al., 2018). Truly transdisciplinary research requires elevating the role of stakeholders to that of co-producers, so that they are equal to scientists (Klenk et al., 2015).

A final gap refers to ethical considerations about social data management (Flint et al., 2017), power dynamics, and researcher positionality in fieldwork with participants (Rangecroft et al., 2021). These topics were not mentioned in the reviewed studies. When working with human-related data, researchers must follow FAIR principles (Wilkinson et al., 2016), minimise risks to participants, obtain informed consent, and protect people's privacy (Flint et al., 2017). Privacy concerns are especially important when dealing with sensitive data about people, particularly high-resolution spatial data, consumer data, and digital

trace data from social media (Zipper et al., 2019). Furthermore, there are risks for disadvantaged groups and marginalised minority populations that need to be considered (Kounadi and Leitner, 2014). Hence, socio-hydrologists need to pay more attention to the proper management of social data.

## 5 Conclusions

This article has provided an overview of the state of the art of socio-hydrology studies in the field of natural hazards and

disaster research. The aim was to scrutinise the field's maturity in relation to different aspects. Although considerable achievements have been made during this first decade of socio-hydrology development, our systematic review revealed and re-confirmed many persisting gaps in the areas of natural hazards and disasters, especially regarding the degree to which current approaches are actually holistic.

We conclude that the scholarly debate on what constitutes 'social' in socio-hydrology research is timely and urgent. Our results

showed that hydrology has often overlapped with social sciences with no deep exchange between them and a predominance of hydrology perspectives (Figure 10a). Notwithstanding philosophical, methodological, and communication differences, some studies did indeed apply techniques and methods from both natural and social sciences (Figure 10b). This demonstrates that they can be used complementarily to provide a more holistic perspective about complex problems. However, the study of coupled human-water systems still has a long way to go in terms of integrating several disciplines and stakeholders. Within

this context, we recommend that socio-hydrology should consider social and natural sciences knowledge equally while at the same time involving stakeholders in order to produce new understandings (Figure 10c). An emphasis on linking research to the practical realities of stakeholders is essential to enhance the impact of socio-hydrological studies.


Based on the identified challenges, we highlighted specific research needs that will play an important role in extending socio-hydrology in the coming years and ensuring it is capable of holistically addressing natural hazards and disasters. We expect

this discussion to encourage socio-hydrologists to investigate different types of disasters using a more integrative approach that better combines the natural and social sciences, and by exploring mixed approaches, involving stakeholders, and broadening the use of cross-scale analyses. The consideration of the identified research gaps can help to strengthen socio-hydrology research and enhance its relevance to scientists, practitioners and decision-makers to better support DRR.

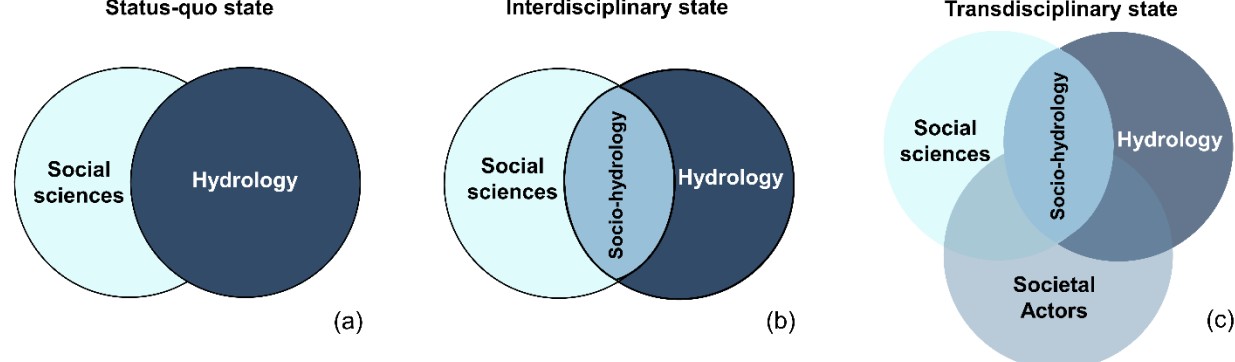

**Figure 10: Conceptual model of socio-hydrology: (a) status-quo state where social sciences and hydrology approaches 'overlap', (b) interdisciplinary state where social scientists and hydrologists interact, and (c) transdisciplinary state where in addition to the interaction of different disciplines, new understandings are produced by interactions with society.**

**Author contributions.** F.M.V contributed to the conceptualization, data curation, investigation, methodology and writing

(original draft preparation, and review and editing). M.K contributed to the project administration, supervision, and writing (review and editing). M.M.B contributed to the conceptualization, methodology, supervision, and writing (review and editing).

**Competing interests.** The authors declare that they have no conflict of interest.

**Acknowledgements.** The first author thanks the Brazilian National Council for Scientific and Technological Development (CNPq) for research scholarship.

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
