# Peer review of "To which extent are socio-hydrology studies truly integrative? The case of natural hazards and disaster research"

_Hydrology and Earth System Sciences, 2021_

## Author Response (AR1)

Dear Editor,

Thank you for conducting this manuscript evaluation.

We report in this document our response to each one of the referee's comments and questions. To aid the visualization of the changes, we submitted a "track changes" version of the manuscript where we`ve highlighted the locations in the text with changes.

Dear Anonymous Referee #1,

We are very grateful for your informative and constructive comments and observations, which surely improve the manuscript. We accepted and incorporated the suggestions into the paper.

**1) "The authors conducted a systematic review of the literature on natural hazards and disaster within sociohydrology. The paper identifies trends and gaps in the literature and proposes some ways to address existing issues. The paper is well written, concise, and clear. It also provides some useful visual aids for those who want to get a glimpse in the current state of the art. I particularly appreciate the proposed shift from an interdisciplinary state to a transdisciplinary state. In my opinion, the paper can be accepted upon minor revision. While I am not necessarily against its publication in HESS, I believe the paper would fit NHESS better, considering its focus, but I leave this to the handling editor."**

Thank you for the careful reading and the positive appreciation of our manuscript. Regarding the publication in HESS, we believe that the special issue *Contributions of transdisciplinary approaches to hydrology and water resources management* from *Hydrology and Earth System Sciences* has the ideal set of readers to our findings. Thus we are very sure that the submission of this manuscript to this special issue is perfectly appropriate.

**2) "I would leave percentages out of the abstract (not wrong to have them, just a style suggestion)"**

We agreed with this comment and removed the percentages of the abstract. The abstract now reads as follows:

*"Abstract. Given the recent developments in socio-hydrology and its potential contributions to disaster risk reduction (DRR), we conducted a systematic literature review of socio-hydrological studies aiming to identify persisting gaps and discuss tractable approaches for tackling them. A total of 44 articles that address natural hazards or disasters were reviewed in detail. Our results indicated that: (i) Most of the studies addressed floods whereas there were very few research applications for droughts and compound or multi-hazards; (ii) none of the articles investigated interactions across temporal and spatial scales; (iii) quantitative approaches were used more often in comparison to mixed and qualitative approaches; (iv) monodisciplinary studies prevailed over multi or interdisciplinary ones, and (v) a reduced number of the articles involved stakeholder participation. In summary, we found that there is a fragmentation in the field, with a multitude of social and physical components, methods and data sources being used. Based on these findings, we point out potential ways of tackling the identified challenges to advance socio-hydrology, including studying multiple hazards in a joint framework and exploiting new methods for integrating results from qualitative and quantitative analyses to leverage on the strengths of different fields of knowledge. Addressing these challenges will improve our understanding of human-water interactions to support DRR."*

**3) "The authors use the affiliation as a proxy for the discipline to which a certain author belongs. I understand the reasoning behind this and I agree this is one way of doing it. However, nowadays an increasing number of researchers find themselves employed by a department which doesn't (partially or fully) reflect their expertise, especially with the rise of interdisciplinary projects that get funded and the consequent interdisciplinarity of teams where people with very different expertise end up working in the same department. I think a better way to deal with this would be to look at the publication record of each author (but extremely time consuming). I am not saying the authors should do this now, but I would refer to this limitation somewhere in the paper (e.g. in methods)."**

Thank you for raising this important issue. We agreed with this comment and included this limitation in the a new Section "5 Potential study limitations":

*"Third, there is a bias in the classification of the articles into monodisciplinary and interdisciplinary. We considered the author affiliation as a proxy for their discipline. However, nowadays, an increasing number of researchers work in interdisciplinary projects whose affiliation department does not reflect their expertise. Hence, although the present results can be a sufficient indicator of the current disciplinarity scenario, the studies' interdisciplinarity should be investigated when possible. This could be done by, for instance, analysing the publications record of each author."*

**4) "p.9 line 204: it is not entirely clear what is meant with temporal series of social data, could the authors maybe shortly elaborate on that?"**
Thank you for this observation. The correct expression is "time series". We corrected it in the text.

**5) "p.15 line 332: in my opinion, droughts so far have received little attention for two reasons. The first one is that the field of sociohydrology was "initiated" by hydrologist who have mostly focused on floods throughout their career. The second one concerns the characteristic of a drought, e.g. more complex phenomenon, larger spatial scale and longer temporal scale, cross-country impacts, etc., which make it more challenging to study even more so when coupled to the human component. Maybe this can also be briefly discussed in the paper."**
Thank you for raising this point. About the first reason, we did not analyse the previous publications of the socio-hydrologists and we did not have references for underpinning this sentence. We agreed with the second reason and considered it in the topic "4 Research Agenda":

*"The first persisting knowledge gap is related to the predominance of flood studies. Even though droughts and multi-hazard events also cause considerable damages, they received little attention. This result can be explained by the fact that drought and multi-hazard events have complex characteristics, which make their investigation in a coupled system more challenging. For droughts, their onset, cessation and spatial extent are notoriously problematic to determine (de Brito et al., 2020). With regard to multi-hazard events, multiple interconnections must be considered when studying them (Kappes et al., 2012)."*

**6) "p.15 line 334, "all types of disasters can be considered as socio-hydrological phenomena as they are directly and/or indirectly associated with water". I am not sure I agree with this, as I don't see the connection between, for example, earthquakes and water (unless they trigger a tsunami) or volcanic eruptions and water."**
This sentence in the text is a citation from Vanelli and Kobiyama (2021). In their article, the authors explored this subject and discussed that water can be a direct and/or an indirect trigger of most disasters, and, mainly, water is essential in the disaster response to ensure the public health of affected people. In some hazards, like storms and extreme temperature (Gopalakrishnan, 2013), the direct and/or indirect role of water is more clear. However, some geophysical researches indicate that some volcanic eruptions occur due to an increase in water vapor pressure in the subduction zones (Plank et al., 2013; Shaw et al., 2008; Walowski et al., 2015). Water in fault zones can act as a kind of lubricant that enables two adjacent blocks of rocks to move past each other, this movement can give rise to earthquakes (Hubbert and Rubey, 1959). Furthermore, in post-disaster of all types of disasters, the affected people require potable water for consumption and non-potable water to clean and disinfect their houses, avoiding or minimizing diseases (Kouadio et al., 2012; Pan American Health Organization, 2006; Suk et al., 2020; UN News, 2021; Vanelli and Kobiyama, 2021).
We modified the end of the sentence to clarify the meaning:

*"Besides this, all types of disasters can be considered socio-hydrological phenomena because disasters are directly and/or indirectly associated with water, which can act as the triggering agent or/and is indispensable during disaster response (Vanelli and Kobiyama, 2021). Hence it is necessary to advance our understanding of interwoven social and environmental processes by considering the interplay between society and different types and combinations of natural hazards. To this end, new methods and data are needed to consider the dynamics between consecutive and compound hazards and society."*

**7) "Technical corrections:**
**p.2 line 60: the main objective of this study is**
**p.5 line 117: I would cite the article you mention here (I believe Sivapalan, Savenije, and Blöschl 2012?)**

**p.7 line 171: what do the authors mean with "one-person household", is it "one person per household"?**

Thank you for your observations. We corrected all of them.

**References**

Gopalakrishnan, C.: Water and disasters: A review and analysis of policy aspects, Int. J. Water Resour. Dev., 29(2), 250–271, doi:10.1080/07900627.2012.756133, 2013.

Hubbert, M. and Rubey, W. W.: Role of Fluid Pressure in Mechanics of Overthrust Faulting, Bulletin of the Geological Society of America, 70, 115-166, 1959.

Kouadio, I. K., Aljunid, S., Kamigaki, T., Hammad, K. and Oshitani, H.: Infectious diseases following natural disasters: Prevention and control measures, Expert Rev. Anti. Infect. Ther., 10(1), 95–104, doi:10.1586/eri.11.155, 2012.

Pan American Health Organization: The Challenge in Disaster Reduction for the Water and Sanitation Sector: Improving quality of life by reducing vulnerabilities, Pan Am. Health, 49, 2006.

Plank, T., Kelley, K. A., Zimmer, M. M., Hauri, E. H. and Wallace, P. J.: Why do mafic arc magmas contain ~4wt% water on average?, Earth Planet. Sci. Lett., 364, 168–179, doi:10.1016/j.epsl.2012.11.044, 2013.

Shaw, A. M., Hauri, E. H., Fischer, T. P., Hilton, D. R. and Kelley, K. A.: Hydrogen isotopes in Mariana arc melt inclusions: Implications for subduction dehydration and the deep-Earth water cycle, Earth Planet. Sci. Lett., 275(1–2), 138–145, doi:10.1016/j.epsl.2008.08.015, 2008.

Suk, J. E., Vaughan, E. C., Cook, R. G. and Semenza, J. C.: Natural disasters and infectious disease in Europe: A literature review to identify cascading risk pathways, Eur. J. Public Health, 30(5), 928–935, doi:10.1093/eurpub/ckz111, 2020.

UN News. United Nations Global perspective Human stories. Haiti earthquake: Waterborne disease poses new threat to children. Available on: <https://news.un.org/en/story/2021/09/1099122>. 2021.

Vanelli, F. M. and Kobiyama, M.: How can socio-hydrology contribute to natural disaster risk reduction?, Hydrol. Sci. J., 66(12), 1758–1766, doi:10.1080/02626667.2021.1967356, 2021.

Walowski, K. J., Wallace, P. J., Hauri, E. H., Wada, I. and Clynne, M. A.: Slab melting beneath the Cascade Arc driven by dehydration of altered oceanic peridotite, Nat. Geosci., 8(5), 404–408, doi:10.1038/ngeo2417, 2015.

Dear Maurizio Mazzoleni

We are very grateful for your review. Please find our response to each one of your comments and questions below.

1) **"This study aims at performing a systematic literature review of socio-hydrological studies for identifying the main gaps and discussing a research agenda for addressing them. The paper deals with a timely and important topic. I have really enjoyed reading this paper, and I believe it provides a significant contribution to the field of socio-hydrology. However, I do have a number of comments before the paper can be considered for publication. Below are my main concerns."**

Thank you for the careful reading of our manuscript. We appreciate the constructive feedback. In this document, we answer each comment and question.

2) **"It is not really clear to me the procedure used for the selection of the 44 papers. I understand that the focus of the study is on natural hazards (i.e. floods and droughts). However, why agriculture, water consumption, and groundwater were used as exclusion criteria? Those are crucial components for drought impact and have a great influence on human-water dynamics. For example, Garcia et al. (2016) showed that drought awareness can significantly shape per-capita water demand, which in turn affects water use, reservoir volume, and consequent water shortage during drought periods. Similar examples can be found in Van Emmerik et al (2014) and Gonzales and Ajami (2017), which were not included in your review. Why did you decide to neglect these components of interdisciplinarity and key aspects in human-water dynamics? They may not explicitly include the term "socio-hydrology" but they are definitely important studies for unraveling human-water dynamics, i.e. the focus of socio-hydrology. Including more studies will definitely strengthen your review."**

We agree with you that the more comprehensive a review is, the better their outcomes tend to be. However, we consider the following points:

(1) in a systematic review, a clear and sometimes narrow research scope needs to be defined in order to have a reasonable number of articles that can be manually read. A search in the Web of Science database using the term "human-water" and "socio-hydrology" leads to 700 and 310 results, respectively. Including all these articles using a traditional systematic review approach would not be feasible. To address this gap, future studies could apply systematic mapping or bibliometric analyses. These techniques allow addressing a larger number of articles as suggested by the reviewer. However, they tend not to provide a deep and more detailed overview of the studies addressed.

(2) recently other systematic reviews were performed concerning socio-hydrology in general, for instance, Fischer et al. (2021) and Herrera-Franco et al. (2021). Yet, even though, recent studies (e.g. Di Baldassarre et al., 2018; Vanelli and Kobiyama, 2021) discussed the potential contributions of socio-hydrology to study disaster risk reduction, no review has specifically focused on natural hazards, risks and disasters.

(3) although socio-hydrology focuses on human-water dynamics, not at all studies in the general literature about human-water dynamics assume socio-hydrology as the viewpoint. In some cases, perspectives other than socio-hydrology can be applied (e.g. socio-ecological system, nexus approach).

With these 3 points in mind, we decided to focus our research on socio-hydrological studies in the fields of natural hazards, risks and disasters. Thus, the starting point was the definition of socio-hydrology as the search term. In the following step, we defined inclusion criteria to achieve our goal: understanding the relation between socio-hydrology and natural hazards, risks and disasters research. In this context, we included only socio-hydrological studies that explicitly mentioned natural hazards, risks, or disasters. This decision was based on the complexity of the coupled human-water system because all processes are interlinked and interconnected. The rationale for decisions taken as well as complete documentation of the reviewed articles are provided in the manuscript (See Section "2 Methods" and Supplementary Material).

We are aware that we might miss some relevant articles that did not fit to inclusion criteria. This is a common problem with any systematic review (Ivanova et al., 2020). Therefore, we added a new Section about the study's limitations ("5 Potential study limitations"). We also added the references you suggested as examples of relevant articles that deal with natural hazards but did not mention our search terms and inclusion criteria. The modified text read as follows:

*"5 Potential study limitations*
*While the present study provided an inter and transdisciplinary account of barriers in socio-hydrological research applied to natural hazards, risks and disasters, some caveats should be considered when interpreting the obtained results. First, although we used a comprehensive set of keywords, we may have missed relevant articles during the screening process of review (Figure 1). For instance, (Van Emmerik et al., (2014), Garcia et al., (2016), and Gonzales and Ajami, (2017) address aspects related to natural hazards and awareness, which are relevant for understanding socio-hydrological phenomena. This is a common problem with any systematic review, as researchers risk missing important references given the language, search terms, and inclusion criteria used (Ivanova et al., 2020). Meanwhile, despite their drawbacks, systematic review procedures provide a deep and more detailed overview of the studies addressed than other techniques like systematic mapping or bibliometric analyses.*
*Second, we focused only on articles that mention socio-hydrology in the title, abstract, and/or author's keywords. However, studies that deal with understanding human and water interactions without mentioning socio-hydrology could contribute to a deep understanding of how these interactions are considered in disaster and risk research. In future studies, besides socio-hydrological studies related to natural hazards, risks and disasters, human-water dynamics studies can be reviewed aiming to analyse and compare the methods used. It can be interesting to compare how human-water interactions are addressed through different lenses (e.g. nexus approach, socio-ecological system, complexity theory).*
*Third, there is a bias in the classification of the articles into monodisciplinary and interdisciplinary. We considered the author affiliation as a proxy for their discipline. However, nowadays, an increasing number of researchers work in interdisciplinary projects whose affiliation department does not reflect their expertise. Hence, although the present results can be a sufficient indicator of the current disciplinarity scenario, the studies' interdisciplinarity should be investigated when possible. This could be done by, for instance, analysing the publications record of each author.*

*Nevertheless, despite these potential limitations, the present study is the first to present a systematic review of socio-hydrological studies applied to natural hazards, risks and disasters. The sample of included articles provides sufficient information to stimulate discussion aiming to address challenges in this field of knowledge."*

**3) "Have you considered the option to include studies that deal with understanding and modeling human-water dynamics without being considered as socio-hydrological studies? For example, Haer et al. (2020)? It would be really interesting to compare these studies with socio-hydrological studies to discuss differences in both physical and social methods used to represent human-water dynamics."**
We agree with your comment and consider that many articles are not called "socio-hydrological" studies but could be considered such. However, for reasons delineated in comment 2, manually reviewing all human-water dynamic studies (about 700 articles only in the Web of Science database) would not be feasible. In addition, not all human-water dynamic studies assume a socio-hydrology perspective because they can consider other lenses, for instance, socio-ecological systems, coupled human and natural systems (CHANS), or nexus approach. Thus, we had to define a clear focus, which is understanding the current state of the art regarding socio-hydrological studies in natural hazards, risks and disasters research. The suggested comparison is interesting and meaningful. We added it as a recommendation for future studies in the new above-mentioned Section: "5 Potential study limitations". See the modified text below:

*"Second, we focused only on articles that mention socio-hydrology in the title, abstract, and/or author's keywords. However, studies that deal with understanding human and water interactions without mentioning socio-hydrology could contribute to a deep understanding of how these interactions are considered in disaster and risk research. In future studies, besides socio-hydrological studies related to natural hazards, risks and disasters, human-water dynamics studies can be reviewed aiming to analyse and compare the methods used. It can be interesting to compare how human-water interactions are addressed through different lenses (e.g. nexus approach, socio-ecological system, complexity theory)."*

**4) "Based on the commentary of Di Baldassarre et al. (2021), multiple approaches should be implemented for better understanding and representing human-water dynamics. In your review, you compared a number of categories (e.g. type of natural hazard investigated, spatial and temporal scales of the social and physical systems, physical and social components, etc.). Would it be beneficial for this review also to include a comparison between socio-hydrological studies using quantitative vs qualitative observations?"**
We are not sure if we understood your question correctly. One of our study's main points is indeed to analyse quantitative and qualitative data gathering sources and processing methods. Section "3.5 Trends regarding the methods used to understand coupled social and physical systems" is dedicated to this discussion.

**5) "Line 241: What do you mean by "calibrate societal memory"? In the models you cited, awareness variations influence demography, which in turn affect flood losses, and not the other way around. Also, calibration in modeling applications has a different meaning than the one you are referring to. I suggest you modify the sentence accordingly in order to avoid misunderstandings."**
Thank you for raising this point. We modified the sentence as follow in order to clarify the text:

*"Among these studies, 34.1% calculated "Societal memory/Risk awareness/Risk perception" based on the proportion of flood damage, assuming that the individual memory is a function of disaster's exposure (e.g. Albertini et al., 2020; Di Baldassarre et al., 2013; Buarque et al., 2020)."*

**6) "Line 276: use "system dynamic" modeling rather than the term "stylized model". Why are you considering that only Sugeng et al. (2019) used system dynamic modeling? Also Di Baldassarre et al. (2013, 2015) used system dynamics model. I suggest that you define the different types of modeling techniques at the beginning of this section. The fact that Sugeng et al. (2019) used Vensim software (i.e. stock and flow representation) does not mean that the other socio-hydrological studies did not use system dynamics as they used differential equations, which are equivalent to the stocks and flows formulation in vensim."**
We performed the classification based on the explicit information written by the article's authors. In this regard, Sugeng et al. (2019) explicitly mentioned the use of system dynamic modelling, while Di Baldassarre et al. (2013, 2015) do not mention this term. Nevertheless, we agree with your comment. Thus, we removed the sentence above-mentioned and the term "stylised model". Given that all studies included in the review are illustrations or representations of the dynamics of the coupled human and water systems,

we removed the class "System dynamic" aiming to avoid misunderstandings. We modified the graphs and included these articles in a new category called "Empirical numerical modelling".

**7) "Line 377: Are you sure that socio-hydrology uses the same methods and perspectives as traditional hydrology? In my opinion, approaches like system dynamics and ABM were just recently introduced for socio-hydrological applications"**

Thanks for the comment. We did not affirm that socio-hydrology uses the same methods and perspectives as traditional hydrology. Instead, we posed a question with the intent of stimulating reflection on the topic. Based on our systematic review results, we identified several gaps and found that indeed, socio-hydrology research tends to be monodisciplinary. We thus ask the reader "If socio-hydrology uses the same methods and perspectives as traditional hydrology, can we expect it to deliver different and new insights into complex human-water systems?". This question aims to encourage researchers in a critical viewpoint. We modified the sentence to clarify our point. If the editor deems the question too contradictory, we can remove it entirely.

Furthermore, system dynamics and agent-based models are techniques well-established before socio-hydrology development and they are used in a range of disciplines other than socio-hydrology. For instance, NetLogo is a multi-agent programmable modeling environment, where several sample models of different disciplines can be found, including hydrological studies.

"*This raises the question for reflection: if socio-hydrology uses the same methods and perspectives as traditional hydrology, can we expect it to deliver different and new insights into complex human-water systems?*"

**References**

Di Baldassarre, G., Viglione, A., Carr, G., Kuil, L., Salinas, J. L. and Blöschl, G.: Socio-hydrology: Conceptualising human-flood interactions, Hydrol. Earth Syst. Sci., 17(8), 3295–3303, doi:10.5194/hess-17-3295-2013, 2013.

Di Baldassarre, G., Viglione, A., Carr, G., Kuil, L., Yan, K., Brandimarte, L. and Blöschl, G.: Debates - Perspectives on socio-hydrology: Capturing feedbacks between physical and social processes, Water Resour. Res., 51(6), 4770–4781, doi:10.1002/2014WR016416, 2015.

Di Baldassarre, G., Nohrstedt, D., Mård, J., Burchardt, S., Albin, C., Bondesson, S., Breinl, K., Deegan, F. M., Fuentes, D., Lopez, M. G., Granberg, M., Nyberg, L., Nyman, M. R., Rhodes, E., Troll, V., Young, S., Walch, C. and Parker, C. F.: An Integrative Research Framework to Unravel the Interplay of Natural Hazards and Vulnerabilities, Earth's Futur., 6(3), 305–310, doi:10.1002/2017EF000764, 2018.

Fischer, A., Miller, J. A., Nottingham, E., Wiederstein, T., Krueger, L. J., Perez-Quesada, G., Hutchinson, S. L. and Sanderson, M. R.: A Systematic Review of Spatial-Temporal Scale Issues in Sociohydrology, Front. Water, 3(September), 1–19, doi:10.3389/frwa.2021.730169, 2021.

Herrera-Franco, G., Montalván-Burbano, N., Carrión-Mero, P. and Bravo-Montero, Lady: Worldwide research on socio-hydrology: A bibliometric analysis, Water (Switzerland), 13(9), 1–28, doi:10.3390/w13091283, 2021.

Ivanova, D., Barrett, J., Wiedenhofer, D., Macura, B., Callaghan, M. and Creutzig, F.: Quantifying the potential for climate change mitigation of consumption options, Environ. Res. Lett., 15(9), doi:10.1088/1748-9326/ab8589, 2020.

Sugeng, E. S., Varma, N. and Smith, Z. A.: Evaluation of the Normalisasi Policy in Jakarta, Indonesia Using System Dynamics, Landsc. Archit. Front., 7(3), 78, doi:10.15302/j-laf-1-020005, 2019.

Vanelli, F. M. and Kobiyama, M.: How can socio-hydrology contribute to natural disaster risk reduction?, Hydrol. Sci. J., 66(12), 1758–1766, doi:10.1080/02626667.2021.1967356, 2021.